# Prediction of Deep Myometrial Infiltration, Clinical Risk Category, Histological Type, and Lymphovascular Space Invasion in Women with Endometrial Cancer Based on Clinical and T2-Weighted MRI Radiomic Features

**DOI:** 10.3390/cancers15082209

**Published:** 2023-04-08

**Authors:** Xingfeng Li, Michele Dessi, Diana Marcus, James Russell, Eric O. Aboagye, Laura Burney Ellis, Alexander Sheeka, Won-Ho Edward Park, Nishat Bharwani, Sadaf Ghaem-Maghami, Andrea G. Rockall

**Affiliations:** 1Department of Surgery and Cancer, Imperial College Hammersmith Campus, Du Cane Road, London W12 0NN, UK; 2Chelsea and Westminster Hospital, 369 Fulham Rd., London SW10 9NH, UK; 3The Imaging Department, Imperial College Healthcare NHS Trust, UK Hammersmith Hospital, Du Cane Road, London W12 0HS, UK

**Keywords:** endometrial cancer, T2-weighted MRI, radiomics, machine learning classification, feature selection, deep myometrial infiltration, clinical risk score, histological type, lymphovascular space invasion

## Abstract

**Simple Summary:**

Deep myometrial infiltration, clinical risk score, histological type, and lymphovascular space invasion are important clinical variables that have significant management implications for endometrial cancer patients. Determination of these factors using pure T2-weighted MRI is time-consuming, and the accuracy of this relies on the experience of the clinicians. Combining clinical information and radiomic features from MRI, we developed machine learning classification models to predict these clinical variables. Based on a training dataset, an automatic selection classification model with an optimized hyperparameters method was adopted to find the optimal classifiers. The accuracy of the model predictions was evaluated using an independent external testing dataset. The results suggest that an integrated model (combining clinical and radiomic features) achieved a reasonable accuracy for endometrial cancer clinical variable prediction. The application of these models in clinical practice could potentially lead to cost reductions and personalized treatment.

**Abstract:**

Purpose: To predict deep myometrial infiltration (DMI), clinical risk category, histological type, and lymphovascular space invasion (LVSI) in women with endometrial cancer using machine learning classification methods based on clinical and image signatures from T2-weighted MR images. Methods: A training dataset containing 413 patients and an independent testing dataset consisting of 82 cases were employed in this retrospective study. Manual segmentation of the whole tumor volume on sagittal T2-weighted MRI was performed. Clinical and radiomic features were extracted to predict: (i) DMI of endometrial cancer patients, (ii) endometrial cancer clinical high-risk level, (iii) histological subtype of tumor, and (iv) presence of LVSI. A classification model with different automatically selected hyperparameter values was created. The area under the curve (AUC) of the receiver operating characteristic (ROC) curve, F1 score, average recall, and average precision were calculated to evaluate different models. Results: Based on the independent external testing dataset, the AUCs for DMI, high-risk endometrial cancer, endometrial histological type, and LVSI classification were 0.79, 0.82, 0.91, and 0.85, respectively. The corresponding 95% confidence intervals (CI) of the AUCs were [0.69, 0.89], [0.75, 0.91], [0.83, 0.97], and [0.77, 0.93], respectively. Conclusion: It is possible to classify endometrial cancer DMI, risk, histology type, and LVSI using different machine learning methods.

## 1. Introduction

Endometrial cancer is the sixth most common malignant disorder worldwide with an estimated 382,000 new cases diagnosed annually [1]. Magnetic resonance imaging (MRI) has been extensively applied in endometrial cancer staging [2,3,4,5] and is usually the imaging modality of choice for assessment of deep myometrial invasion (DMI) [6,7,8,9]. Studies have shown that using MRI alone, endometrial cancer DMI prediction [10] accuracies were estimated to be 74% [11], 76% [12], and 89% [13]. However, these accuracies are subjective as it is dependent on the radiologist’s ability to detect subtle imaging textures on MR images. To overcome these limitations, radiomics [14,15], which extracts quantitative image features from medical imaging, has been proposed to predict DMI [8,9,16]. Nevertheless, relatively few studies to date have focused on identifying clinical prognostic factors other than stage, using radiomic features from MRI [17]. Endometrial cancer histological subtype, International Federation of Gynecology and Obstetrics (FIGO) stage, disease grade, presence of lymphovascular space invasion (LVSI), and DMI are all well-established prognostic biomarkers in endometrial cancer [16,18]. Prediction of histological type pre-operatively would enable patients to be informed of their likely prognosis earlier, and allow for optimization of treatment planning [19].

Thus far, studies using radiomics for DMI classification prediction [8,10,20] have applied only one machine learning method, which may not represent the optimal predictive ability of machine learning. For example, a recent study [8] adopted a random forest to assess DMI, whilst a second study [21] applied a logistic regression method. Other studies used a support vector machine (SVM) method [20] and logistic regression [22] for risk stratification. These studies did not compare different machine learning methods to achieve the best predictive results. Therefore, these models may be suboptimal.

Finally, most of the recent endometrial cancer classification studies utilized small- (n < 100) [10,17] or medium-sized (100 ≤ n < 200) [8,20] patient cohorts, which may introduce bias because the small dataset is likely to produce a significantly overoptimistic performance for machine-learning-based estimates (>50%) [23]. As a result, these inaccuracies could lead to significant treatment differences when applied in a clinical environment.

The aim of our study was to develop and validate machine learning models that can accurately predict DMI, risk category, histological subtype [17], and presence of LVSI in a cohort of 495 endometrial cancer patients from multiple centers, attempted through radiomic analysis of preoperative T2-weighted MR images of the primary endometrial tumor.

## 2. Materials and Methods

The aim of this study was to develop and test classification models that will later be further validated as part of a larger prospective study (ClinicalTrials.gov NCT03543215, https://clinicaltrials.gov/ (accessed on 5 April 2023). This study followed the radiomics quality score guidance [14] and the transparent reporting of a multivariable prediction model for individual prognosis or diagnosis guidelines [24]. Patients were identified through an institutional tumor board database between 2007 and 2019 from 15 UK hospitals.

### 2.1. Training and Testing Datasets

Initially, 611 cases were recruited as training datasets; the inclusion criteria regarding MRI were replicated from a previous study [25]. For this classification study, the specific inclusion criteria regarding clinical data were: (1) availability of DMI (cancer stage IA vs. IB) information, information on LVSI, histological risk score, and histological type, and (2) no other type of co-existing cancer. After applying these criteria, 413 cases were included as the training dataset.

Turbo spin echo sequences were implemented to collect the T2-weighted MRI data for 410 training cases (the sequence information of 3 cases was missing). The median value for the echo time and repetition time was 100 milliseconds and 4100 milliseconds, respectively. The median value of slice thickness, spacing between slices, and image reconstruction matrix was 4 mm, 5 mm, and 512 × 512, respectively. Four subjects were scanned with 3.0 T scanners, while the remaining majority of the 406 subjects were scanned with a 1.5 T scanner. The independent holdout dataset included 102 patients collected from 2017 to 2019, and 82 cases were included in the final analysis once the aforementioned inclusion criteria were applied [25]. Table 1 shows the clinical target/response variable and the number of cases for the classification studies.

### 2.2. Data Analysis Pipeline

The image and data analysis pipeline is displayed in Figure 1. A medical image segmentation software (ITK-snap, version 3.6.0, http://www.itksnap.org, 5 April 2023) was employed to delineate each tumor region slice by slice from the T2-weighted MR image by four radiologists (JR, AS, NB, and AR) with a minimum of four years’ experience reading uterine MRI [25,26]. The segmented images were saved as a mask image for analysis in the later steps. An example of the segmentation result is displayed in Figure 2. The radiologists were blinded to the outcome measure.

For T2-weighted MRI, an N4 toolbox (https://github.com/ANTsX/ANTs/wiki/N4BiasFieldCorrection, accessed on 5 April 2023) was applied to correct for the nonuniformity in MRI caused by magnetic field inhomogeneity [27]. For all T2-weighted MRI resolutions were obtained from the Neuroimaging Informatics Technology Initiative (Nifti) header file. To minimize the effect of image resample, the median value of all T2-weighted MRI was used as the final resampled image resolution, i.e., 0.625 mm × 0.625 mm × 5 mm in this study. The image resample was implemented with Simple-ITK (version 5.3, https://simpleitk.org/, accessed on 5 April 2023). For the MRI image, the cubic spine was adopted for the interpolation, while for the associated binary mask image, nearest neighbor interpolation was applied. To avoid different scanner effects, T2-weighted images were normalized using a Z-score method [25]. Image feature extraction was performed using TexLAB tool (version 2.0) on MATLAB (version R2022b; The MathWorks Inc., Natick, MA, USA; http://www.mathworks.com/, accessed on 5 April 2023), pyRadiomics (version 3.0.1, https://www.radiomics.io/pyradiomics.html, accessed on 5 April 2023), and Scikit-image (version 0.19.3, https://scikit-image.org/, accessed on 5 April 2023) in Python (Python3.8, https://www.python.org/, accessed on 5 April 2023). Correlated features (i.e., where the correlation coefficient was larger than 0.99) obtained from these three toolboxes were removed. Eventually, 2168 qualitative and quantitative radiomic features were obtained from T2-weighted MRI and associated mask images. Radiomic and clinical features (age at diagnosis, grade, stage, and risk score) were used, and a machine learning model was developed to predict the target variable DMI, clinical high-risk, histological type, and LVSI.

### 2.3. Patient DMI

The presence of DMI determines the stage of the patient; the absence of DMI is classified as stage 1A, and the presence of DMI whilst the tumor is confined to the uterus is stage 1B. Patients with cancer stages 2 or above were excluded from the analysis in the creation of the predictive model. In total, 292 training cases and 60 holdout cases were included in the DMI predictive study (Table 1).

For patient DMI prediction, because DMI is associated with cancer stage and therefore with cancer risk score, these two clinical variables were not included as predictors in the model for the DMI classification study.

### 2.4. Cohort Treatment and Clinical Risk Classification

All endometrial cancer patients underwent surgical hysterectomy as the routine standard of care. The specimens were analyzed by dedicated gynecological histopathologists per standard protocol using the 2009 FIGO staging system [18]. Patients were subcategorized into binary low and high-risk clinical categories based on their radiological stage and histological grade, as has been described in a previous study [28] and as shown in Table 2. Risk classification was mainly based on cancer stage, and two steps were included in the classification. A simplified version of this classification system was used in the first step, with patients classified into one of 4 clinical risk score groups: low, intermediate, high, and advanced (Table 2). The second step was to combine the intermediate, high, and advanced risk groups into one high-risk group. Patients in the low-risk category had a radiological stage 1A tumor with endometrioid histology grade 1 or 2. All other patients were classified as high risk [28].

### 2.5. Statistical Analysis

The mean and standard deviation of each feature from the training dataset were calculated. A Z-score method (https://uk.mathworks.com/help/stats/zscore.html, accessed on 5 April 2023) was applied to normalize each feature (i.e., remove the mean value and then divide its standard deviation) for the training dataset. The mean and standard deviation values were then applied to normalize the external testing dataset.

Statistical analysis and machine learning classification models were established using MATLAB (R2022b). Specifically, an automatically selected classification model with optimized hyperparameter function (fitcauto.m, https://uk.mathworks.com/help/stats/fitcauto.html, accessed on 5 April 2023) and an elastic net regularization for generalized linear models (LASSO, glmlasso.m https://uk.mathworks.com/help/stats/lassoglm.html, accessed on 5 April 2023) in MATLAB Statistics and Machine Learning Toolbox (Version 12.4) were implemented to build classification models. In the model determined by the LASSO method, the intercept was included in the model in addition to the selected features. In Figure 1, fitcauto was used to denote the method using fitcauto.m function in MATLAB and LASSO to represent the method of glmlasso.m (as the LASSO method is a linear method, and it can be classified into logistic regression classifier). For both methods, predictors included all radiomic features and age at diagnosis. For the fitcauto method, clinical categorical variables such as cancer grade, cancer stage, and risk score were included adaptively for target variable classification; the fitcauto method selects the classification model with optimized hyperparameters automatically and simultaneously. Before using the method, a univariate feature ranking for classification using chi-square tests (fscchi2.m, https://uk.mathworks.com/help/stats/fscchi2.html?searchHighlight=fscchi2&s_tid=srchtitle_fscchi2_1, accessed on 5 April 2023) was adopted to select the eight most important features in the model.

The LASSO method selects features from elastic net regularization for generalized linear models. To apply this method for classification studies, a binormal distribution was adopted, and the maximum number of features was set to be 8 in this study. A 10-fold cross-validation was implemented. With the exception of age at diagnosis, other clinical variables such as grade, stage, and risk score were not included as predictors in the LASSO method, because the current version of the LASSO method from MATLAB does not provide the functionality to handle categorical variables.

The accuracy of the machine learning classification model was evaluated using different statistical measurements. Specifically, after the best model associated with the parameter was determined, the area under the curve (AUC) of the receiver operating characteristic (ROC) curve [29] (AUC for short thereafter), average precision (AP), average recall (AR), F1 score [30] were applied to assess the model prediction accuracy based on testing datasets. Different accuracy measurements such as the F1 score were included because the data were not balanced (see Table 1). The AUC and confidence intervals (CIs) were calculated using a bootstrap method with 1000 repetitions.

## 3. Results

Based on the models obtained from the training datasets and using the fitcauto and LASSO methods, these results were from external testing data with response variables of DMI, clinical risk, histological type, and LVSI.

### 3.1. DMI Prediction Results

From the fitcauto results, the compact SVM with the linear kernel was selected as the best classifier. The eight most important features (See Appendix A) were included in the model. Based on the SVM classifier model, accuracy measurements from the testing data were displayed in Figure 3A (ROC curve), Figure 3B (precision–recall curve), and Figure 3C (confusion matrix). The AUC was 0.79 (95% CI was [0.69,0.89]) (Figure 3A). The data were imbalanced; there were 46 cases of stage IA endometrial cancer patients and there were 14 cases (class 1 in confusion matrix as shown in Figure 3C) with cancer stage IB (DMI cases). Figure 3B shows the corresponding precision–recall curve. The F1 was 0.7, the average recall value was 0.67, and the average precision was 0.72.

Similarly, the LASSO method was applied to select features and determine the model. Seven radiomic features were selected in the model (for details of these features, see Features for DMI classification in Appendix A). The ROC curve (Figure 3D), the precision–recall curve (Figure 3E), and the confusion matrix (Figure 3F) were plotted. The accuracy measurements such as AUC and F1 values were displayed in these figures. All classification accuracy measures are smaller with the LASSO method, in comparison to the fitcauto method (comparing Figure 3A–C with Figure 3D–F).

### 3.2. Clinical Risk Classification

In this experiment, the fitcauto method selected the compact classification ensemble classifier as the best classification model. The results of the ROC curve (Figure 4A,D), the precision–recall curve (Figure 4B,E), and the confusion matrix (Figure 4C,F) were presented. The most important eight selected predictors were plotted in the section of features for risk classification in the Appendix A, where another set of features employed in the LASSO method was also included. The AUC for the classification with the fitcauto method was 0.84 (95% CI was [0.75, 0.91]) (Figure 4A). For the LASSO method, the AUC accuracy to predict clinical risk classification was lower than the fitcauto method (AUC = 0.67 in Figure 4D). The F1 value in Figure 4B (0.72) is also larger than the F1 value in Figure 4E (0.59).

### 3.3. Histological Type Classification

The purpose of endometrial cancer histological type classification was to distinguish whether a tumor had an endometroid or alternative histology. The optimal method determined by fitcauto was a compact classification ensemble classifier. Clinical variables such as cancer stage, risk score and radiomic variables were included in the model selection, and the univariable selection method found eight variables. Cancer grade was not included in the model selection because this information is currently only diagnosable with a biopsy, where histological results would also be available. The results are shown in Figure 5A (ROC curve), 5B (precision–recall curve), and 5C (confusion matrix). The AUC was 0.91 (95% CI [0.833, 0.966]), and the F1 was 0.72 despite the unbalanced dataset. The included features for the fitcauto and LASSO methods were presented (Appendix A in the feature for histological type classification of Appendix A). The results from LASSO are given in Figure 4D (ROC curve), Figure 4E (precision–recall curve), and Figure 4F (confusion matrix). Compared with the ensemble method, except for the average precision measurement (AP is 0.96 for the LASSO method, Figure 5E), the LASSO method is inferior to the ensemble method in other accuracy measurements. Furthermore, the CI from the LASSO method is larger than the results from fitcauto (comparing the cloud in Figure 4A with Figure 4D).

If cancer grade was included as a predictor in the classification model for fitcauto method, the accuracy of the model for histological prediction improved significantly (AUC of 0.97). Detailed results can be found in the Appendix A.

### 3.4. LVSI Classification

In this experiment, the fitcauto method selected the compact classification ensemble classifier as the best approach. Figure 6 shows the ROC (Figure 6A) and precision–recall curves (Figure 6B), and confusion matrix (Figure 6C) from a classification kernel model determined by the fitcauto method. The classification kernel is a trained model object for a binary Gaussian kernel classification model, which uses random feature expansion. The selected features included cancer grade, cancer stage, and risk (see the bar plot in Appendix A). The AUC value for the classification was 0.85 with a 95% CI of [0.77, 0.93] (Figure 6A).

In addition, the LASSO method was applied to select predictors from age and radiomic features. The classification accuracies were computed. The ROC curve (Figure 6D), precision–recall (Figure 6E), and confusion matrix (Figure 6F) are displayed in Figure 6. The full list of selected features is presented in the Appendix A.

## 4. Discussion

We have developed, compared, and tested machine learning models to predict DMI, clinical risk stratification, histological type, and LVSI for patients with endometrial cancer. For machine learning predictions, we found that no single classification method could accomplish the best accuracy for all response variables. This is because most of the machine learning algorithms were developed based on different assumptions; the machine learning methods achieve the best results if the inputted data meet these assumptions. It could also be due to potentially inappropriate hyperparameters of the machine learning model because these parameters can affect the accuracy outcomes of the model when predicting different target variables. In short, to achieve clinically relevant predictions, various models with differing numbers of radiomic and clinical features need to be evaluated. We found that the predictive results improved if we combined both clinical and radiomic features. Therefore, it is necessary to use both clinical and radiomic features to best translate the model to clinical practice.

Given the accuracy in predicting DMI in this study, clinical use of this machine learning algorithm may aid in determining clinical management, such as referral to a cancer center, and simple prognostication, allowing for early intervention with alternate or additional therapies [31,32,33]. In a previous study based on the random forest classifier [10], an AUC of 0.94 was achieved using a testing dataset. However, based on our data with the SVM classifier, we only obtained 0.79 of AUC. The main difference is that the previous study only had 54 cases [10]. Additionally, they normalized the features from training and testing datasets in the same way, which is not realistic or applicable to real-life datasets. Moreover, they used a weighted accuracy on unbalanced data that can falsely amplify the accuracy. Other studies [8,9] obtained a DMI classification accuracy of AUC (0.68) and AUC (0.81) from their testing dataset, which are closer to our results. However, although these included medium-sized cohorts, these studies employed a study dataset of less than 200 patients. More importantly, the testing data of these studies were obtained from the same source as the training dataset, while our dataset is collected independently of the training dataset.

For clinical high-risk prediction, a previous study showed an AUC accuracy of 0.76 from the testing data for risk classification [9]. If cancer grade is included in our risk model, a classification AUC of 0.84 (Figure 4A) can be achieved; however, using LASSO with only radiomic features, the classification accuracy model results in an AUC of 0.67 (Figure 4D). Another recent study [22] showed the low-risk predictive model with an AUC accuracy of 0.74. This accuracy is close to our LASSO results; this is because both methods used the logistic regression method for the classification.

The histological subtype of endometrial cancer was predicted with a high accuracy (AUC of 0.91) (Figure 5A). The clinical application of the model would be to assist with decision-making for adjuvant treatment, as non-endometrioid cancers can benefit from treatment beyond surgery [19]. Comparing Figure 5A with Figure 5D, the CI of AUC (cloud in the figures) from fitcauto is smaller than the results from LASSO. Although the data is unbalanced, the F1 (0.86) is also high (Figure 5B). The classification accuracy measurement obtained from the LASSO method results were inferior to the classification kernel method for histological subtype prediction. This is because the model did not include the clinical categorical variables such as cancer grade and stage, which are important predictors for the classification. For the histological subtype prediction, clinical categorical variables of cancer grade were not included for model selection, because cancer grade information was obtained from the biopsy. If we include cancer grade in the model, the classification accuracy in terms of AUC can reach 0.97 (Appendix A), suggesting that cancer grade is an important predictor for the classification.

LVSI has been repeatedly shown to be an independent negative prognostic factor for recurrence and survival [34]. LVSI is a histological diagnosis, made by confirmation of tumor emboli within micro vessels. It has been shown to be a strong prognostic factor in recurrence and lymph node metastasis [34,35], but is not perceptible to the naked eye on MR images. Accurate prediction of LVSI involvement pre-operatively using MR images, as observed in this radiomic study, can allow for early adjuvant intervention and reduce the chance of repeat surgery to confirm lymph node spread. The current study was able to predict the presence of LVSI with an AUC of 0.85 from the testing dataset (Figure 6A), which is comparable to a recent study that obtained an accuracy of AUC of 0.80 using features from multi-sequence MR images [8]. Another study [36] achieved an AUC of 0.82 with multiparametric radiomic features and clinical features. Our study achieved similar results by adopting radiomic features from sagittal T2-weighted images and clinical features, which would incur a lower cost for data collection and implementation.

There are several limitations to this study. Firstly, although we applied univariate model selection and LASSO methods for the feature selection, other methods such as principal component analysis for feature reduction could be applied and compared. Secondly, we included 495 endometrial cancer patients in this study. Further work is needed to upgrade these models to achieve better discrimination accuracy with even larger datasets. Finally, most current studies employed only AUC to assess the classification models [8,9,10,20,22], while AUC has several flaws [37] such as it being sensitive to class imbalance [38]. In this study, we included the F1 score and precision–recall curve in our analysis. The F1 score can be used to prevent the target/response variable imbalance problem. Shortly, to improve our model’s evaluation, we employed AUC, average precision, average recall, and F1 score criteria. However, other metrics such as the Matthews correlation coefficient [30,38] should be studied as the endometrial cancer data are often imbalanced.

## 5. Conclusions

In conclusion, our study shows that good accuracies can be achieved using larger datasets (>200 patients) for predicting clinical response variables of endometrial cancer using T2-weighted MRI. There appears to be no single optimal machine learning classification method to achieve the best results for all clinical target variables for predicting clinical and pathological features of endometrial cancer using clinical and T2-weighted MRI data. Our results suggest that for different outcomes, the optimal classification method and associated hyperparameters need to be selected. Our model also had a high prediction accuracy for the histological subtype (AUC = 0.91 using testing data) and a reasonable prediction accuracy of patient DMI, risk, and LVSI. Prospective validation of this model will be required with larger datasets.

## Figures and Tables

**Figure 1 cancers-15-02209-f001:**
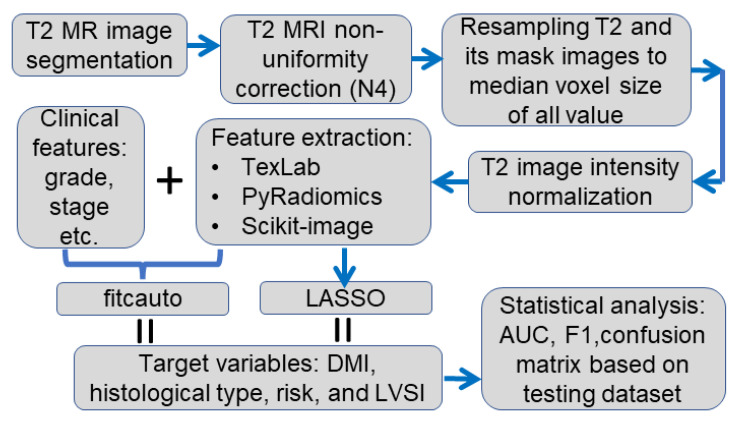
Data analysis pipeline. The arrows indicate the order of the analysis. fitcauto represents automatically selected classification model with optimized hyperparameters using fitcauto.m function in MATLAB. LASSO: least absolute shrinkage and selection operator; AUC: area under the ROC curve.

**Figure 2 cancers-15-02209-f002:**
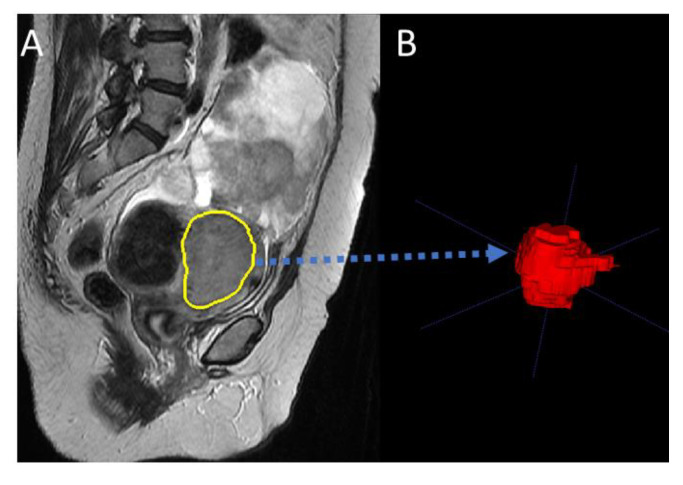
An example of manual tumor segmentation. (**A**) T2-weighted image, the yellow curve shows the location of the tumor; (**B**) segmented tumor mask region in 3D.

**Figure 3 cancers-15-02209-f003:**
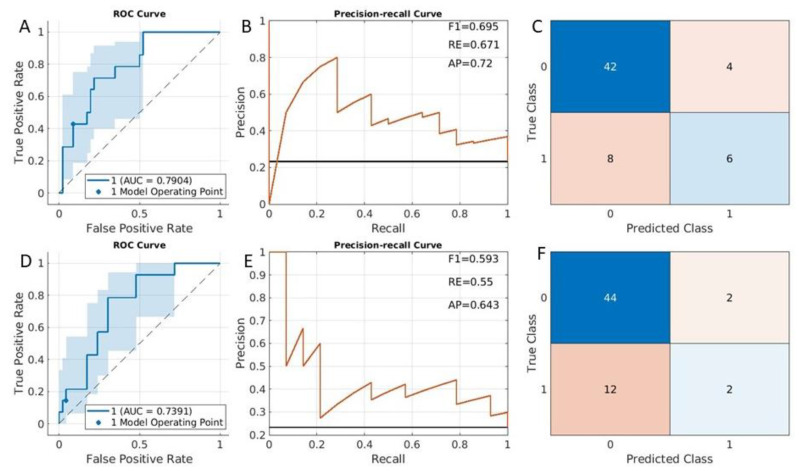
Deep myometrial infiltration prediction results from testing datasets. Summary receiver operating characteristic curves (**A**,**D**), precision–recall curve (**B**,**E**), and confusion matrix (**C**,**F**). (**A**–**C**) were obtained from the fitcauto method, while (**D**–**F**) were from the LASSO method. RE: average recall value. AP: average precision.

**Figure 4 cancers-15-02209-f004:**
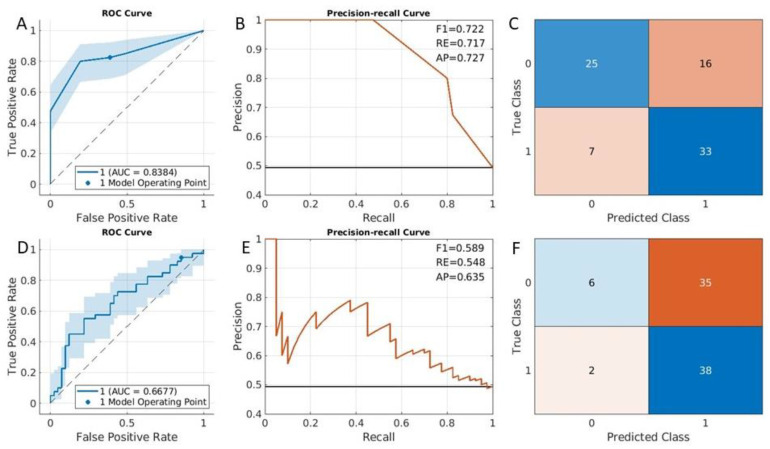
Clinical risk category classification results from testing dataset. ROC curve (**A**), precision–recall curve (**B**) and confusion matrix (**C**) were computed from an ensemble method. ROC curve (**D**), precision–recall curve (**E**), and confusion matrix (**F**) were obtained from LASSO method.

**Figure 5 cancers-15-02209-f005:**
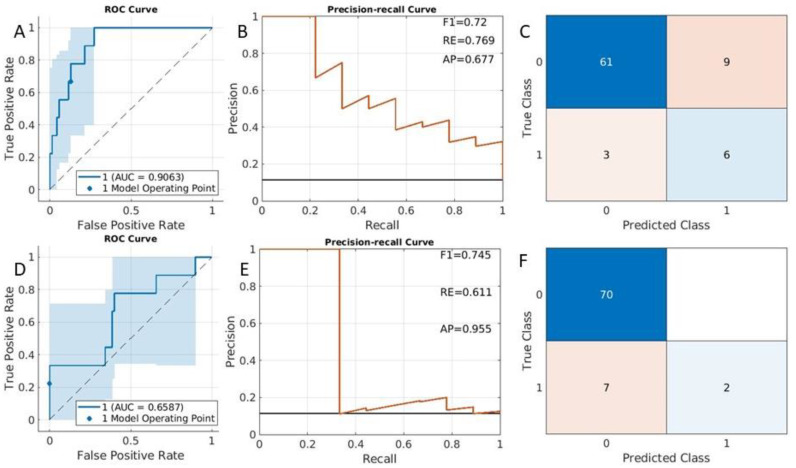
Cancer histological type classification results from testing datasets. (**A**,**D**) are ROC curves; (**B**,**E**) are precision–recall curves; (**C**,**F**) are confusion matrices. (**A**–**C**) were obtained from an ensemble classification method, while (**D**–**F**) were obtained from the LASSO method.

**Figure 6 cancers-15-02209-f006:**
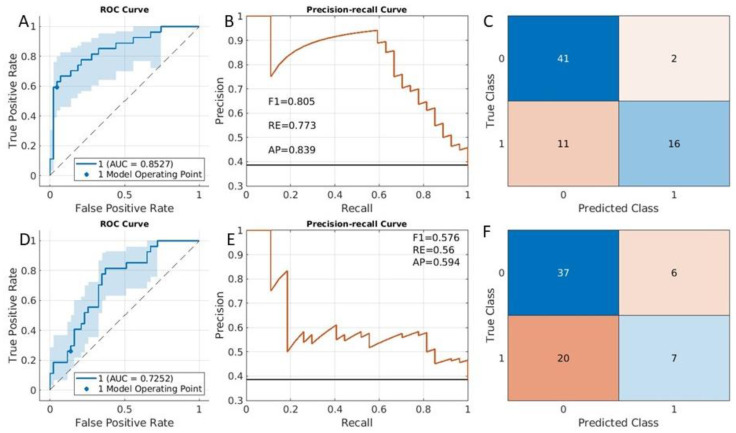
Lymphovascular space invasion prediction from testing datasets. ROC curves (**A**,**D**), precision–recall curves (**B**,**E**). confusion matrix (**C**,**F**). (**A**–**C**) were obtained from classification kernel classifier determined by the fitcauto method. (**D**–**F**) were obtained from the LASSO method.

**Table 1 cancers-15-02209-t001:** Training and testing data for target/response variables. Four binary target/response variables, i.e., DMI, clinical risk, histological type, and presence of LVSI were included as rows; the columns of the training and testing show the number of subjects used in the machine learning classification study.

Target Variable	Training (n)	Training (%)	Testing (n)	Testing (%)
DMI				
no DMI	199	68%	46	76%
DMI	93	32%	14	24%
Clinical risk				
Low	150	36%	41	50%
High	263	64%	40	50%
Histological type				
Endometrioid	301	73%	70	89%
Other types	111	27%	9	11%
LVSI				
Positive	141	36%	27	39%
Negative	248	64%	43	61%

**Table 2 cancers-15-02209-t002:** Endometrial cancer patient risk group classification.

Risk Classification	Criteria
Low	Stage 1A endometrioid grade 1–2
Intermediate	Stage 1A endometrioid grade 3 Stage IB (Grade 1 and Grade 2) with endometrioid type
High	Stage IB Grade 3 endometrioid type Stage 2 endometrioid type Stage 1 or 2 non-endometrioid type.
Advanced	Stage 3 or 4 any type

## Data Availability

Data are unavailable due to privacy or ethical restrictions.

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
