# Peer review of "Prediction of Deep Myometrial Infiltration, Clinical Risk Category, Histological Type, and Lymphovascular Space Invasion in Women with Endometrial Cancer Based on Clinical and T2-Weighted MRI Radiomic Features"

_cancers, 2023, doi:10.3390/cancers15082209_

Round 1
Reviewer 1 Report
The radiomic evaluation was obtained only on T2 w images. I think the results can be improved if the authors add the evaluation of ADC maps.
Author Response
Thank you for this comment. We agree. ADC is a quantitative measurement which is unfortunately not available on all our MRI scanners. In our cohort, we have some subjects who have only a small number of subjects who have been scanned with (diffusion weighted imaging (DWI). As a result, we did not include features from ADC maps in the study.
Reviewer 2 Report
The authors have conducted the study to develop and compare different classification models to classify endometrial cancer based on key features.
1. The authors noted in their conclusion – “It is possible to classify endometrial cancer DMI, risk, histology type, and LVSI using different machine learning methods.” But this is known from prior studies. It appears that other approaches individually, have been taken with different datasets in other studies. In this manuscript some more combinations of models have been evaluated. Authors have done robust evaluation of the predictive models. The results show that the output in terms of AUC for accuracy is quite similar to earlier studies for many of the categories, though the authors claim they got better results due to combination of clinical and radiomic features that was not done for other studies. Please clarify what is the key incremental finding in this study in conclusion.
Author Response
Thank you for this comment. As we have pointed out in the introduction section of the manuscript, firstly, previous studies employed only a small number of endometrial cancer patients, which can lead to a larger estimation bias. Although our classification accuracies are similar to previous studies, these results from previous studies have not been verified in larger cohorts. Secondly, these studies did not compare different machine learning methods for the classification study, whereas in our study we have compared different methods in order to identify the best machine learning method. Thirdly, previous studies used multi-sequences MRI with a small number of radiomic features, which is expensive to collect the MRI , and not practical in terms of clinical use. Additionally, some previous studies have used their testing and training datasets from the same cohort, which leads to an over-estimation of accuracy. Lastly, previous studies have not investigated endometrial cancer histological type prediction using radiomic features. We have developed an integrated model and a model from radiomic features for the histological type classification.
To avoid repetition, we showed the text in boldface in the manuscript. Also, we have added a sentence to the conclusion to help to highlight this further:
In conclusion, our study shows that good accuracies can be achieved using larger datasets (>200 patients) for predicting clinical response variables of endometrial cancer using T2-weighted MRI.’
Reviewer 3 Report
Dear Authors,
thanks for the interesting paper.
I've just some comment to address.
- Why did you chose to segment the tumors only in the sagittal images? I guess because tumors were more obvious in this plane, but segmentation also in other planes may improve data conspicuity (although at the risk of redundancy).
- I know the paper is about T2w imaging (since T2w imaging is still the best imaging modality for cervical cancer), but have you also considered to include other sequences in the model, like DWI? Please add a comment.
- Please, add in text how you manage different sequences/scanners/parameters. In Figure 1 is reported a resample processing step, but need to be specified also in the text.
- At line 94, please correct spin echo to turbo spin echo (I guess sequences were TSE, otherwise you may still scanning the patients)
- At line 95, please report TR and TE in ms rather than seconds.
Thanks again for the interesting paper.
Best regards
Author Response
- Why did you chose to segment the tumors only in the sagittal images? I guess because tumors were more obvious in this plane, but segmentation also in other planes may improve data conspicuity (although at the risk of redundancy).
We chose sagittal images for the segmentation as this provides the optimal view of the tumor from the MRI images. Also, endometrial cancer is best examined in the sagittal plane, providing longitudinal views of the uterus and cervix as well as surrounding structures such as bladder, rectum, and loops of bowel (https://www.ncbi.nlm.nih.gov/pmc/articles/PMC6636928/ ). For these reasons, we chose sagittal MRI for the segmentation.
- I know the paper is about T2w imaging (since T2w imaging is still the best imaging modality for cervical cancer), but have you also considered to include other sequences in the model, like DWI? Please add a comment.
Thank you for this comment. We agree. Some of our subjects were scanned with non-DWI scanners, so we were unable to include this in our study, unless we significantly reduced the size of the cohort. This is the main reason we did not include ADC maps in the study. However, even without the features from ADC map, our classification prediction results are similar or better than the current methods which adopted ADC features for the classification.
- Please, add in text how you manage different sequences/scanners/parameters. In Figure 1 is reported a resample processing step, but need to be specified also in the text.
All T2-weighted MRI resolutions were obtained from the Neuroimaging Informatics Technology Initiative (Nifti) header file. To minimize the effect of image resampling, the median value of all T2-weighted MRI was used as final resampled image resolution, i.e., 0.625mmx0.625mmx5mm in this study. The image resample were implemented with Simple-ITK (version 5.3, https://simpleitk.org/ ). For MRI image, the cubic spine was adopted for the interpolation, while for the associated binary mask image, nearest neighbor interpolation was applied. To avoid difference scanner effect, T2-weighted images were normalized using a Z-score method.
We added these in the manuscript.
- At line 94, please correct spin echo to turbo spin echo (I guess sequences were TSE, otherwise you may still scanning the patients)
Ok. We added turbo in front of the sentence.
- At line 95, please report TR and TE in ms rather than seconds.
Ok. Seconds were replaced by milliseconds.
Reviewer 4 Report
Despite its complexity, I found the present work pleasant to read. It deals with an interesting topic and is well-presented. The study has several strengths (e.g., well-detailed methodology, good and diverse sample size, multiple approaches tested, presenting confusion matrices). However, I have the following comments:
1) Making a model holistic integrating clinical variables is wise. However, it seems that there might be an overlap between the data to predict and the clinical data provided to the algorithms, which might at least partly explain why clinical variables are dominating in terms of importance. In this light, does radiomics significantly add value to the predictive models? If we were to use only clinical and conventional imaging data, would the predictive performance be significantly lower?
2) Do the authors think that there might be an overlap between clinical data provided to the model and prediction aim (e.g., risk includes histological type -I guess at biospy- and is provided to the model to predict histological type - at surgery I suppose-)? Could this represent some sort of data leakage in the authors opinion? Should discrepancies between biopsy and final histology be analyzed in this setting?
3) References 11 and 13 are quite outdated and the authors might want to consider presenting more recent studies and/or meta-analysis showing the accuracy metrics
4) Where would the authors place the presented models in the diagnostic pathway? What would be their clinical significance? Could the authors further discuss on this? Why would the added work and complexity of radiomics be valuable?
5) Ref 30 is mentioned as "a previous study" but this are guidelines and it would be better to acknowledge them as such. Also, I think it would be useful if authors could comprehensively describe the definitions of the clinical variables and how they were obtained as well as reference standard and prediction goal items.
Author Response
1) Making a model holistic integrating clinical variables is wise. However, it seems that there might be an overlap between the data to predict and the clinical data provided to the algorithms, which might at least partly explain why clinical variables are dominating in terms of importance. In this light, does radiomics significantly add value to the predictive models? If we were to use only clinical and conventional imaging data, would the predictive performance be significantly lower?
We carried out additional statistical tests for these questions using holdout data (testing dataset), i.e., we applied testcholdout.m from MATLAB (https://uk.mathworks.com/help/stats/testcholdout.html#bupt6wj_sep_shared-p) to test the predictive performance of these models.
We compared integrated model (using both clinical and radiomic features), clinical model (using cancer risk as a predictor only), and radiomic model (using features from MRI image only). Based on testing dataset, the classification accuracy was 0.85, 0.82, and 0.80 for integrated model, radiomic model, and clinical model respectively. We did not find significant difference between models, i.e., we did not reject the null hypothesis at the Alpha (5%) significance level. Based on the testing data, we obtained:
Integrated model vs clinical model with risk as a predictor only (p =0.11),
Clinical model with risk as a predictor only vs radiomic model (p =0.64)
Integrated model vs radiomic model (p=0.34)
In brief, we did not find significant difference between these models, suggesting that the integrated model improves the estimation accuracy, although it did not reach a significant level.
2) Do the authors think that there might be an overlap between clinical data provided to the model and prediction aim (e.g., risk includes histological type -I guess at biospy- and is provided to the model to predict histological type - at surgery I suppose-)? Could this represent some sort of data leakage in the authors opinion? Should discrepancies between biopsy and final histology be analyzed in this setting?
We do think there might be overlap between clinical data and radiomic features, but the relation between the response variable (prediction aim) and the predictors (clinical data) is complex and difficult to model. The purposes of our study or other classification studies were to map/model the relation between predictor and response variable. In our study, we also included the prediction results using radiomic features only (See results from LASSO method). The results are reasonable (AUC is about 0.7) which suggests that we can predict the response variable using by using MRI image alone, without even checking other aspects of the patients.
We agree that biopsy and finial histology need to be analyzed to confirm and validate the model. We are unable to include these results in this study at this stage, as it is not the main purpose of the study.
3) References 11 and 13 are quite outdated and the authors might want to consider presenting more recent studies and/or meta-analysis showing the accuracy metrics
We replaced reference 11 and 13 with the following references in the manuscript:
The DMI accuracy for the classification was 74% (published in 2021) in:
https://obgyn.onlinelibrary.wiley.com/doi/10.1111/aogs.14146
and DMI with accuracy of 89% based on experts (published in 2020):
https://www.ncbi.nlm.nih.gov/pmc/articles/PMC7197887/
4) Where would the authors place the presented models in the diagnostic pathway? What would be their clinical significance? Could the authors further discuss on this? Why would the added work and complexity of radiomics be valuable?
There are studies to implement radiomics into clinical diagnostic studies (e.g., https://pubs.rsna.org/doi/10.1148/rg.2021210037, https://eprints.lincoln.ac.uk/id/eprint/43205/7/Fournier2021_Article_IncorporatingRadiomicsIntoClin.pdf )
To implement the models from our study, once the T2-weighted MRI is acquired, the first 4 steps for image analysis in Figure 1 need to be carried out. As we have the model, we do not need to extract all features, but we only need to obtain the features in the models with the software tools. After that we predict the outcomes with the model and the extracted features within the model.
Radiomics studies have been applied within oncology to assist radiologists to improve diagnosis accuracy, prognostication, and clinical decision support, with the goal of delivering precision medicine. The models developed in our study have potential to assist radiologist diagnosis, and to be used in pre-treatment diagnosis and prognosis. The models developed in this study are clinically relevant; accurate prediction of DMI would allow the correct patients to be managed in district general hospitals (stage 1a), and for those in whom it is needed, to be operated on in a cancer centre, where they will receive sentinel lymph node assessment in addition to a hysterectomy. Accurate prediction of histological type prior to surgery would ensure the correct patients receive omental biopsies as part of their surgical treatment.
5) Ref 30 is mentioned as "a previous study" but this are guidelines and it would be better to acknowledge them as such. Also, I think it would be useful if authors could comprehensively describe the definitions of the clinical variables and how they were obtained as well as reference standard and prediction goal items.
We add Table 2 to explain the risk classification response variable: A simplified version of this classification system was used, with patients classified into one of 4 clinical risk score groups: low, intermediate, high and advanced (Table 2).
Table 2: Endometrial cancer patient risk group classification
Risk classification |
Criteria |
Low |
Stage 1A endometrioid grade 1–2 |
Intermediate |
Stage 1A endometrioid grade 3 Stage IB (Grade 1 and Grade 2) with endometrioid type |
High |
Stage IB Grade 3 endometrioid type Stage 2 endometrioid type Stage 1 or 2 non-endometrioid type. |
Advanced |
Stage 3 or 4 any type |
The high-risk group includes intermediate, high, and advanced groups as shown in Table 2.
Round 2
Reviewer 4 Report
The Authors satisfactorioly addressed my comments and I have no additional remarks